# Automatic facial expressions, gaze direction and head movements generation of a virtual agent

ALICE DELBOSC, Aix-Marseille Université, France
MAGALIE OCHS, Aix-Marseille Université, France
STEPHANE AYACHE, Aix-Marseille Université, France

In this article, we present two models to jointly and automatically generate the head, facial and gaze movements of a virtual agent from acoustic speech features. Two architectures are explored: a Generative Adversarial Network and an Adversarial Encoder-Decoder. Head movements and gaze orientation are generated as 3D coordinates, while facial expressions are generated using action units based on the facial action coding system. A large corpus of almost 4 hours of videos, involving 89 different speakers is used to train our models. We extract the speech and visual features automatically from these videos using existing tools. The evaluation of these models is conducted objectively with measures such as density evaluation and a visualisation from PCA reduction, as well as subjectively through a users perceptive study. Our proposed methodology shows that on 15 seconds sequences, encoder-decoder architecture drastically improves the perception of generated behaviours in two criteria: the coordination with speech and the naturalness. Our code can be found in : https://github.com/aldelb/non-verbal-behaviours-generation.

CCS Concepts: • **Computing methodologies** → **Neural networks**; **Animation**.

Additional Key Words and Phrases: Non-verbal behaviour, behaviour generation, embodied conversational agent, neural networks, adversarial learning, encoder-decoder

**ACM Reference Format:**
Alice Delbosc, Magalie Ochs, and Stephane Ayache. 2022. Automatic facial expressions, gaze direction and head movements generation of a virtual agent. In *INTERNATIONAL CONFERENCE ON MULTIMODAL INTERACTION (ICMI '22 Companion), November 7–11, 2022, Bengaluru, India.* ACM, New York, NY, USA, 15 pages. https://doi.org/10.1145/3536220.3558806

## 1 INTRODUCTION

Behaviour generation is an active and recent research area. Virtual agents are becoming essential in many applications such as games or virtual environments. To communicate and fully engage humans in the interaction, the non-verbal behaviour of the embodied conversational agent is essential. In human-human interaction, Munhall et al. [28] showed that the rhythmic beat of head movements increases speech intelligibility. Studies have also demonstrated that head movements could increase the level of warmth, competence and improve the way a virtual agent is perceived in general [2, 22]. In the same way, Tinwell et al. [38] showed that "uncanniness" is increased for a character with a perceived lack of facial expressions.

Traditionally, the generation of an agent's body movements and facial expressions requires the intervention of an animator who designs manually believable movements. This work is costly and time-consuming. The approaches based on motion capture remain limited given the costly hardware and the time-consuming post-processing. Automatic generation tools would allow to automate this process and decrease the cost of animation.

*ICMI '22 Companion, November 7–11, 2022, Bengaluru, India*
© 2022 Association for Computing Machinery.
ACM ISBN 978-1-4503-9389-8/22/11...$15.00
https://doi.org/10.1145/3536220.3558806

Seminal works have shown a strong correlation between individual's speech and her/his non-verbal behaviour [5, 17, 24]. Based on these research works, some systems for generating behaviours from speech began to emerge [13, 18, 36, 42]. These models generate behaviours from certain acoustic or textual features extracting from the speech. However, this generation task presents many difficulties. The real challenge is to represent the diversity of the facial and head movements. Indeed, many similar gestures and expressions can plausibly be associated with the same input speech. In a human-human interaction, the perception of a speech expressed by raising the right eyebrow will be perceived in an extremely similar way to the same speech expressed by raising the left eyebrow.

In this paper, we focus on the generation of non-verbal behaviours from acoustic speech features without considering other elements that may influence the behaviour (e.g. social attitudes such as persuasion, personality, communicative style, etc.). As a first step, we do not consider the speech content. We concentrate on the objective to generate believable face, head and gaze movements considering the acoustic features of the speech. Our approach generates automatically and simultaneously head movements, gaze orientation and facial expressions. We propose to explore the performances of two different models for this specific task of behaviour generation: a Generative Adversarial Network (GAN), and an Adversarial Encoder-Decoder (AED). For the sake of reproducibility, all the tools used are open-source, all our code is available on github[1] (with a detailed procedure) and the survey completed by the participants for the subjective evaluation is available online[2].

The paper is organised as follows. After formulating the learning problem (Section 2), we review existing works (Section 3). Then, in Section 4, we present the corpus used and the post-processing performed on the data. In section 5, we present the trained models and in Section 6, we introduce our evaluation method and our results.

## 2 PROBLEM FORMULATION

Our goal is to generate believable non-verbal behaviours of a virtual agent automatically from the acoustic features of the speech given as input. Considering the measures of performance, we aim at identifying a good balance between the accuracy of the model, coverage and diversity of the generated behaviour. It implies generating a set of behaviours as close as possible to the set of possible and diverse human behaviours.

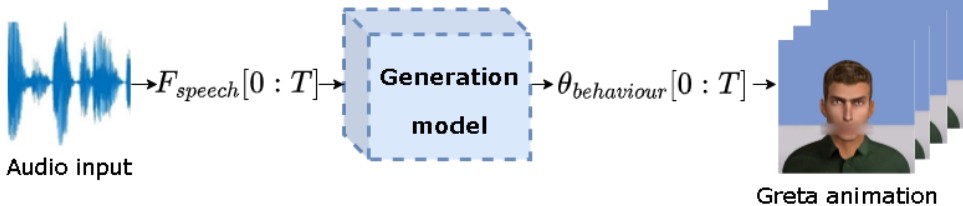

Fig. 1. Behaviour generation process

The problem can be formulated as follows: given a sequence of acoustic speech features $F_{speech}[0:T]$ extracted from a segment of audio input at regular intervals $t$, the task is to generate the sequence of corresponding movements and expressions $\theta_{behaviour}[0:T]$ that a virtual agent should play while speaking. $\theta_{behaviour}[0:T]$ groups $\theta_{head}[0:T]$, $\theta_{gaze}[0:T]$ and $\theta_{AU}[0:T]$, respectively head movements, gaze orientation and facial expressions. The head movements $\theta_{head}[0:T]$ and gaze orientation $\theta_{gaze}[0:T]$ are expressed in 3D coordinates, while the facial expressions $\theta_{AU}[0:T]$ are described by action units (AUs) based on the Facial Action Coding

---

[1]https://github.com/aldelb/non-verbal-behaviours-generation
[2]https://forms.gle/RHcupwN69Po892rJ6

System (FACS) [8]. The notations presented above will be used throughout this article. The figure 1 illustrates this behaviour generation process.

## 3 STATE OF THE ART

The research works on behaviour generation can be described by different characteristics: the approach (rules-based or data-driven), the generation task (types of generated gesture), the characteristics of the corpus, the inputs and outputs of the model, etc. In order to structure the state of art, in the section 3.1, we present examples of rules-based models; in Section 3.2, we describe data-driven models including machine learning models; in Section 3.3, we discuss the outputs representation of the models, and in Section 3.4, we detail the type of corpus considered in previous works. Finally, we summarise selected very recent works in Table 1 to have an overview and compared our work to the characteristics of existing models.

### 3.1 Rules-based systems

The first approach explored for the automatic generation of virtual character's behaviour was based on sets of rules. The rules described the mapping of words or speech features to a facial expression or movements. Cassell [4] and Cao et al. [3] showed that body movements and facial expressions can be synchronised with audio using a set of predefined rules. Marsella et al. [23] and Lhommet et al. [20] developed rules-based systems to generate body movements by analysing the content of the audio input. These approaches limit the generated expressions and movements to a dictionary. However, facial expressions and head movements are based on much more than a limited set of rules. Consequently, after a while, the movements of the virtual character may appear repetitive. Furthermore, this approach is time-consuming to implement because of the temporal synchronisation between speech and gestures to specify in the system. Finally, such methods rely on language-specific rules and do not easily handle multiple languages or multiple speech styles. To overcome these problems, data-driven approaches have been explored more recently.

### 3.2 Data-driven approaches

Data-driven approaches do not depend on experts in animation and linguistics. These approaches learn the relationships between speech and movements or facial expressions. Mariooryad and Busso [22] proposed to replace rules with Dynamic Bayesian Networks (DBN). In Chiu and Marsella [6], a Gaussian Process Latent Variable Models (GPLVM) has been used to learn a low-dimensional layer and select the most likely movements given the speech as input. Levine et al. [19] used Hidden Markov Models (HMM) to select the most likely gesture based on speech. However, these research works are still based on an animation dictionary, limiting the diversity of the generated movements. Moreover, in these models, there is only one motion sequence for an input audio signal. It supports the hypothesis that the speech-to-motion correspondence is injective but the correspondence between acoustic speech features and non-verbal behaviour is a "One-To-Many" problem. For instance, people can tilt the head to one side or to the other while pronouncing the same speech. Given the importance of the variability of the virtual character's non-verbal behaviour, the models of automatic generation should include this diversity of movements.

More recently, deep neural networks shown their superiority in learning from large datasets by generating a sequence of images for the non-verbal behaviour. The generation often focused on head movements or body movements conditioned by a speech input. Many models like normalizing-flow have been used for their generation [15]. Normalizing-flow support only linear operations, limiting the expressiveness of the models [30]. GANs (Generative Adversarial Network) are among the generative models that made the best progress in the last decade [12], in particular conditional GANs [26]. These models can convert acoustic speech features into non-verbal behaviours while preserving the diversity and multiple nature of the generated non-verbal behaviour. Sadoughi

and Busso [35], Takeuchi et al. [37] and Hasegawa et al. [14] used GANs with recurrent neural networks (RNNs). RNNs are used to capture temporal dependencies of the input signal. In particular, they use Bidirectional Long Short Term Memory (B-LSTM) to synthesise body movements from speech. Traum et al. [39] used LSTMs to synthesise head movements from speech. Despite the use of RNNs in previous models, Li et al. [21] shown that convolutional layers are better in the movements generation task, as it prevents the error accumulation, which is specific to RNN.

To reduce the effects of the collapse mode, a very common failure that causes the model to generate only one behaviour, Wu et al. [42] used an Unrolled GAN. If the adversarial training enhances the synchronisation of behaviour with speech, Kucherenko et al. [18] insisted on the importance of a post-processing to smooth the behaviour generated. Another type of model gives good results in generation tasks: Kucherenko et al. [18] proposed an encoder-decoder speech to motion by combining the decoder of a motion auto-encoder, and an encoder which maps speech to motion representations. Habibie et al. [13] used an Adversarial Encoder-Decoder, which means an encoder-decoder combined with adversarial learning. Given the performance of GANs in the area of non-verbal behaviour generation, we choose to adopt a similar approach by exploring and comparing adversarial models.

Most of the previous works only generate facial animations *or* head movements. The generation of facial expressions or head movements presents a different problem. Head movements can be generated in a much more diverse way depending on the subject than facial expressions. However, facial expressions and head movements are all connected and synchronised with speech [5]. As far as we know, only the recent research work of Habibie et al. [13] proposed the automatic generation of facial expressions and head movements jointly from an adversarial approach. In our study, inspired by Habibie et al. [13], we analyse facial expressions and head movements in a combined way, while changing the way facial expressions are represented. Indeed, our work differs from Habibie et al. [13] since we propose to represent facial expressions by explainable features, those are the action units (Section 3.3), and we explore different architectures in an adversarial approach to compare the performances of the models.

## 3.3 Outputs of the models

While body and head movements are always generated with 3D coordinates, facial expressions can be generated in various ways. They can be generated directly with the 3D coordinates of the face, like Karras et al. [16] who used LSTM to learn the 3D coordinates of some key points of the face. Another approach consists in describing these facial expressions using a model, such as Pham et al. [33] who used 3D blendshape face model from the FaceWarehouse database. In our model, we represent the facial expressions using action units (AUs) based on the well-known Facial Action Coding System (FACS). This choice is motivated by the objective to obtain interpretable and explainable results and therefore be able to manipulate the generated facial expressions much more easily than 3D coordinates. Generating action units instead of 3D coordinates presents the main advantage to give us the opportunity to manipulate the output of the model, for instance to adapt the generated action units in order to express particular socio-emotional states like emotions [7, 40]. It's why we consider as particularly important to represent facial expressions with action units.

## 3.4 Corpora

Corpora are required for training and evaluating models with a data-driven approach. In the previous research works, the size of the considered corpora as well as the number of speakers vary: Sadoughi and Busso [35] used a 1h06 corpus with a single speaker, Kucherenko et al. [18] used a 1h51 corpus with two speakers, while Ginosar et al. [11] and Habibie et al. [13] used a 144h corpus with 10 subjects. In these configurations, generated behaviours depend on the styles of the speakers involves in the corpora. In comparison with the state of the art,

we propose in our work a multi-individual adversarial model with 89 distinct speakers, from different ethnic backgrounds.

The table 1 presents a selection of research works presented above that we use as reference. These research works have been selected given their performance in behaviour generation and their proximity in terms of the type of task generation.

| Article | Habibie et al. [13] | Sadoughi and Busso [35] | Kucherenko et al. [18] | Wu et al. [42] |
|---|---|---|---|---|
| Generation task | hand movements, head movements and facial expressions | head movements | body movements | upper body movements |
| Model | Adversarial Encoder-Decoder | CGAN | Encoder-Decoder | CGAN |
| Input signals | MFCC (with derivatives $1^{st}$ and $2^{nd}$) | F0, intensity (with derivatives $1^{st}$ and $2^{nd}$) | MFCC, F0, energy (with derivatives $1^{st}$ and $2^{nd}$) | MFCC, F0, intensity (with derivatives $1^{st}$ and $2^{nd}$) |
| Output signals | 3D coordinates | 3D rotation | 3D coordinates | 3D coordinates |
| Data | 144h with 10 subjects | 1h06 with 1 subject | 1h51 with 2 subjects | 4h57 |
| Evaluation metrics | user studies | user studies and density estimation | user studies and signal comparison in terms of position and speed | user studies, density and speed estimation |

Table 1. Articles that will serve as references

Compared to the state of art, the contributions of the work presented in this paper are: (1) an action unit-based behaviour generation to improve the interpretability of the model outputs: our models jointly generate the head movements, the gaze direction and the facial action units; (2) contrary to existing research works, we propose to consider numerous speakers to cover a wide range of speech styles; (3) to construct these models, we propose two original architectures, inspired by the literature, to compare two data-driven models with an adversarial approach. These models are presented Section 5.

## 4 PRE-PROCESSING OF THE DATA

The lack and quality of data is a major problem for the behaviour generation task. Some methods exist to collect data based on multiple cameras and motion capture systems. However, these methods remain expensive and time-consuming. In this work, we propose to automatically extract the acoustic speech features from an existing corpus using state-of-the-art tools: Openface [1] and Opensmile [9]. Then, the features are aligned to synchronise speech and movements.

*Openface* is a toolkit that detects the head position automatically, gaze orientation and facial action units of a person on a video. The tool extracts features at the frequency of 30 frames per second (30 fps). In our work,

we consider the eye gaze direction represented in world coordinates, the eye gaze direction in radians, the head rotation in radians and the facial action units. We consider the intensity of 17 facial action units from 1 to $5^3$. We obtain a total of 28 features characterising the head, gaze and facial movements. These features, noted $\theta_{behaviour} \in \mathbf{R}^{28}$, are used for the prediction and constitute the output of the generation model. These features are then the input of the animation platform to visualise the generated behaviours. To simulate the behaviours generated by our models on an embodied conversational agent, we use the Greta platform [32].

*Opensmile* is a toolbox that extracts the audio features from a speech. This tool extracts features at a frequency of 50 fps. In this work, we consider the following vocal features commonly used in vocal signal processing: frequency F0 (a global measure of the pitch), shimmer, loudness, and six spectral features (Harmonic difference H1-H2, harmonic difference H1-A3, MFCC 1-4). After features extraction, first and second derivatives of the data are computed and concatenated with other data [13, 42]. In total, we consider 27 vocal features. The vocal features extracted from the human speech are noted $F_{speech} \in \mathbf{R}^{27}$.

Because of the difference of granularity between speech and non-verbal behaviour, a careful alignment of speech and visual features is essential. We perform a resampling to obtain a common alignment on the lowest of the extracted frequencies. We obtain our aligned features at 30 fps.

In terms of the corpus, we use the CMU Multimodal Opinion level Sentiment Intensity (CMU-MOSI) corpus [43]. In this dataset, a speaker discusses a topic in front of the camera, giving her/his opinion about a movie. The speakers themselves filmed these videos, which means that videos are recorded in different setups, sometimes with high-tech cameras and microphones, other times with less professional equipment. There is 89 different speakers from different ethnic backgrounds who expressed themselves in English. In total, we have 92 videos that represent more than 3h58m of recording. We divide this data into training and test sets (we use the test set to validate the hyperparameters of our models), approximately 70-30 respectively lengths of 2h40 and 1h18. We do not check whether the same person appears in both set, although this is unlikely given the number of speakers and the number of videos.

The most widely tested method for the analysis of human behaviour consists in working on short segments of videos (thin-slices) over a sliding window varying from a few seconds to several minutes depending on the socio-emotional phenomena studied [29]. Inspired by this method, the videos in the corpus were cut into 4s segments over a sliding window of 300ms. We then obtain 2437 segments for the training set and 1397 segments for the test set.

## 5  METHODS AND MODELS

Following the research conducted during the state of the art, we implement and compare two different architectures. They both use an adversarial approach. As a result, they are composed of two neural networks: a generator and a discriminator. The generator generates new data and the discriminator have to distinguish the generated data from the real data. The essence of adversarial training is a min-max game between the generator and the discriminator. While the discriminator is optimised to recognise whether an input is generated by the generator or taken from the real data, the generator tries to fool the discriminator by learning how to generate data that looks like the real data. In reality, the generator tries to minimise the Jensen-Shannon divergence between the generated distribution and the real distribution. We recall that the data to be generated are head movements, gaze orientation, and AUs: $\theta_{head}[0:T], \theta_{gaze}[0:T], \theta_{AU}[0:T]$. At the entrance of our models, data are normalised. At the exit, data are smoothed.

---

[3]AU01, AU02, AU04, AU05, AU06, AU07, AU09, AU10, AU12, AU14, AU15, AU17, AU20, AU23, AU25, AU26, AU45.

## Data normalisation

Our two architectures generate a temporal sequence of AUs and movements according to a given speech input. As described in Section 4, the speech is first processed to extract the acoustic features at each time step $t$. The data are then normalised between 0 and 1. This normalisation combined with sigmoid activation layers in the output of our models forces the generated data to be in a range of values determined by our training data. Therefore, the generated data should be close to the reality and we should not obtain totally unbelievable behaviours despite the generation of new behaviours.

## The Architectures

$1^{er}$ **model - DCGAN:** the first architecture is inspired by Wu et al. [42]. We implement a DCGAN (Deep Conditional Generative Adversarial Net). The generator generates data by sampling from a noise distribution (z) and acoustic speech features $F_{speech}[0...T]$. This architecture keeps the randomness of the generated movements. $F_{speech}[0...T]$ plays the role of the condition in the generation. The generator generates a movement conditioned by the audio condition it receives as input. This condition is added to both the generator and the discriminator input. The discriminator measures if the movements look natural, but also if the movements look natural with respect to these audio features and if the temporal alignment is respected. The generator consists of four 1D layers (Conv-BN-ReLU) with kernels of size 3 and MaxPool after every second block. These 1D layers are framed by linear layers and then a sigmoid activation layer. We also add dropout layers after the 1D layers. In a symmetrical way, the discriminator is also made of four 1D layers (Conv-BN-ReLU), linear layers and a sigmoid activation layer. Figure 2 illustrates this architecture.

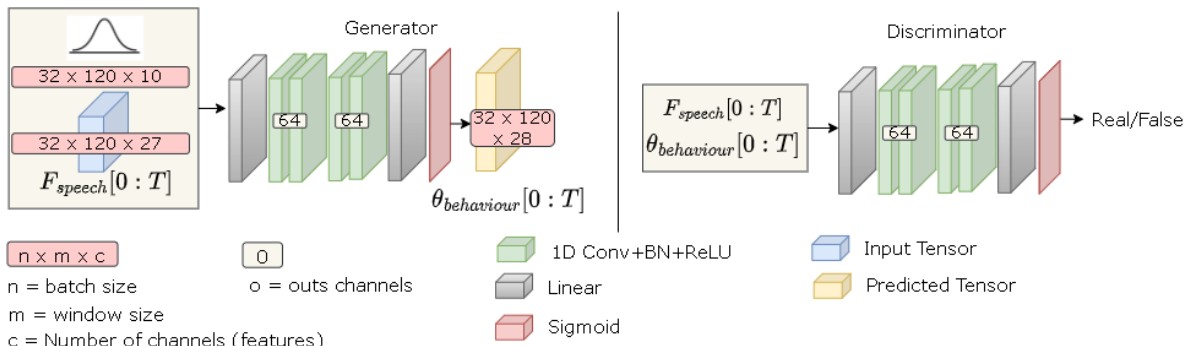

Fig. 2. Architecture of the first model

The collapse mode is a very common failure when training GANs. Once the generator identifies a sample to fool the discriminator, it tends to generate only that sample, regardless of the noise and condition it receives as input. To prevent this failure during training, we implement an unrolled GAN. In GAN, the cost function is computed and then backpropagation is performed to adjust the parameters of the discriminator D and the generator G. In unrolled GAN, the discriminator is trained in the same exact way as the GAN. However to optimise the generator, the model unroll k steps to learn how the discriminator optimise itself for a specific generator. We unroll 10 steps. The unrolling is used by the generator to predict the behaviour, but is not used in the optimisation of the discriminator. We only use the first step to update the discriminator. For the generator, we backpropagate the gradient on all 10 steps [25].

**Training details:** we use Adam for training, with a learning rate of $10^{-5}$ for the generator and the discriminator and a batch size of 32. We train during 1000 epochs. The following equation is for optimising the generator G and the discriminator D.

$$L = \min_G \max_D \mathbb{E}_{F_{speech}}[log(1 - D(F_{speech}, G(z, F_{speech}))] + \mathbb{E}_{F_{speech}, \theta_{behaviour}}[logD(F_{behaviour}, \theta_{behaviour})]$$

The proposed model is inspired by Wu et al. [42]. However, the final architecture differs from Wu et al. [42]. Indeed, we did not use LSTMs but 1D convolution layers. This choice is motivated by the fact that Li et al. [21] have shown that convolutional layers are better in the movements generation task, as it prevents the error accumulation, which is specific to RNN. Moreover, our outputs are not as Wu et al. [42] who use only the 3D coordinates of the upper body movements. In this work, we consider the facial expressions expressed in action units, the head movements and the gaze direction expressed in 3D coordinates.

$2^{nd}$ - **Adversarial Encoder-Decoder:** the second architecture is inspired by Habibie et al. [13]. The generator takes the form of a 1D encoder-decoder. It is an adaptation of the U-Net implementation [34] originally created for 2D image segmentation. This architecture is created to take advantage of the correlation between head movements, gaze orientation and facial expressions. The encoder consists of ten 1D blocks (Conv-BN-ReLU) with size 3 kernels and MaxPool after every second block. Then, three decoders are created symmetrically to generate believable behaviours. Each decoder is associated to a data type with different value intervals: a decoder for head movements, a decoder for eye movements and a decoder for AUs. They consist of seven 1D blocks (conv-BN-ReLU) with kernels of size 3 and UpSampling after every second block. As the decoders are symmetric with the encoder, it uses skip-connectivity with the corresponding layers of the encoder. Figures 3 and 4 illustrate this architecture.

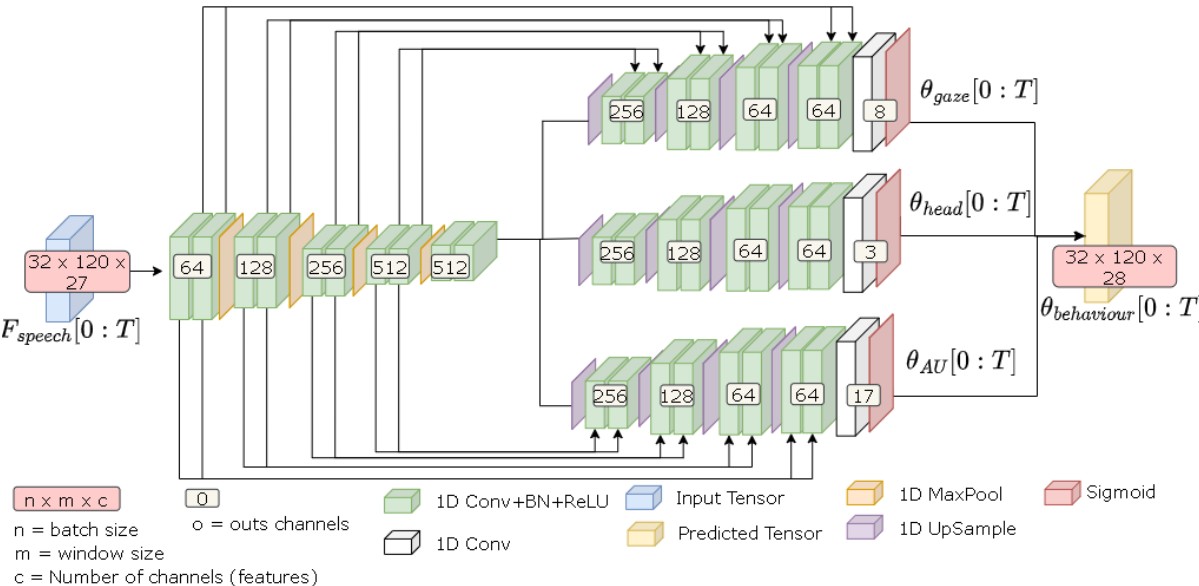

Fig. 3. Generator architecture of the second model

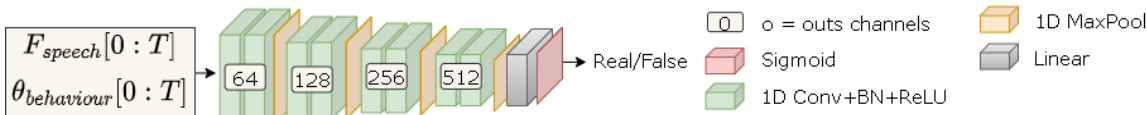

Fig. 4. Discriminator architecture of the second model

As explained in Ginosar et al. [11], to avoid convergence towards the average, ensure believable and expressive behaviours and enhance the synchronisation between behaviours and speech, a discriminator is added to this encoder-decoder to implement adversarial training. As in our previous model, the discriminator must predict whether the generated samples are real or fake. It consists of eight 1D layers (Conv-BN-Relu) with a kernel of size 3 and MaxPool after each second block. Then a linear and sigmoid activation layer. As in the previous architecture, we do not only use the generated behaviours as input, but also the acoustic speech features. We also add dropout layers after the 1D layers.

The authors were inspired by U-Net architecture with a modification of the number of layers and their size. We keep the number of layers and size of the U-Net architecture, transforming the 2D convolution layers into 1D convolution layers. Habibie et al. [13] generated facial expressions and movements in 3D coordinates, unlike them we generate facial expressions using action units.

**Training details:** we use Adam for training, with a learning rate of $10^{-3}$ for the generator and $10^{-5}$ for the discriminator and a mini batch of 32. We train during 1000 epochs. We supervise our generator $G$ with the following loss function:

$$\mathcal{L}_G = \mathcal{L}_{gaze} + \mathcal{L}_{head} + \mathcal{L}_{AU}$$

$\mathcal{L}_{gaze}$, $\mathcal{L}_{head}$ and $\mathcal{L}_{AU}$ are the root mean square errors (RMSEs) of the gaze orientation, head movement, and AUs features.

$$\mathcal{L}_{gaze} = \sum_{t=0}^{T-1} (\theta_{gaze}[t] - \hat{\theta}_{gaze}[t])^2$$

$$\mathcal{L}_{head} = \sum_{t=0}^{T-1} (\theta_{head}[t] - \hat{\theta}_{head}[t])^2$$

$$\mathcal{L}_{AU} = \sum_{t=0}^{T-1} (\theta_{AU}[t] - \hat{\theta}_{AU}[t])^2$$

we pose the adversarial loss function with the discriminator D:

$$L_{adv}(G, D) \quad = \quad \mathbb{E}_{F_{speech}}[log(1 \, - \, D(F_{speech}, G(F_{speech})] \, + \, \mathbb{E}_{F_{speech}, \theta_{behaviour}}[logD(F_{speech}, \theta_{behaviour})]$$

Combining this adversarial loss with the direct supervisory loss, we get:

$$\mathcal{L} = \mathcal{L}_G + w. \min_G \max_D \mathcal{L}_{adv}(G, D)$$

With w set to 0.1 to ensure that each term is equally weighted. In order to compare the results of an encoder-decoder without discriminator, we also analyse $w = 0$ in the section 6.

## Smoothing of data

Visualisation of several generated behaviours played on a virtual agent allowed us to observe that the speed of the generated behaviours, especially those of the head, eyes, eyebrows and mouth, is higher than real behaviours. This phenomenon is observed in particular by Kucherenko et al. [18], our intuition tells us that the number of speakers in our dataset increases it. We therefore perform a post-processing of data, with the Savitzky-Golay algorithm used in signal processing, to smooth our behaviour curves by convolutions, with a polynomial as interpolation function. The parameters of this smoothing are the degree of the polynomial and the number of points to consider. Head movements, gaze direction, mouth and eyebrows are not smoothed with the same parameters. We use a polynomial of degree 7 with a window of 71 points for head movements, a window of 31 points for gaze direction, a window of 21 points for the AUs corresponding to the eyebrows, and finally a window of 11 points for the AUs corresponding to the mouth. To determine these parameters, we empirically and visually evaluate the resulting behaviours on a set of generated videos played on the virtual agent.

The AU corresponding to eyes blink is treated differently. Intermediate values for this AU are not realistic (e.g. an eye half open over several seconds). We assign the maximum value when the model predicts a value higher than the average of the range of possible values and we assign the minimum value when the model predicts a value lower than the average of this range.

Due to the crucial role of smoothing and post-processing, for the final user study we considered only smoothed gestures.

## 6  EVALUATION AND RESULTS

In order to determine the best configurations for our models, we train various models by varying the corpus used (the corpus POM [10] and the corpus MOSI [43]), the considered input features, the normalisation intervals, the loss functions, the type of layers, the dropout, the size of the convolution kernels, the activation functions, the batch size, the learning rates, the number of unrolled step in the Unrolled GAN, or even the parameters of the smoothing function. Then, we evaluate these models using several metrics on the test set to determine the generative models that create the most believable behaviours.

The quality of a behavioural generation model can be assessed using objective measures and/or subjective measures. The objective measures are based on algorithmic approaches and return quantitative values reflecting the performance of the model. The subjective measures are generally based on the evaluation of human observers. Note that since we do not have the same task generation as previous research works, we cannot evaluate our model by comparing our performances to existing models.

## Objective evaluation

First, we define objective metrics. We consider loss functions, usually used in deep learning, and kernel density estimation, used for example by Sadoughi and Busso [35] for the task of behaviour generation. In this paper, we, moreover, propose to explore another objective measure: a visualisation from principal component analysis (PCA) reduction to make an initial assessment of the efficiency of our models.

**The loss function:** during training, by computing the loss on the training set and the test set, we verify that there is no overfitting. Nevertheless, as explained in Wu et al. [42], we cannot select models whose loss function tend to 0. Indeed, this method is not suitable for the task of behaviour generation. In fact, the behaviours may be believable without matching the behaviours in the initial test set. If the initial video raises the right eyebrow, the same effect can be produced by raising the left eyebrow, yet the loss function will result in a high value. This evaluation method also tends to ignore small deviations in behaviour whereas these deviations may have a strong

impact on human perception: for instance, if suddenly the agent brutally balances the head backwards with no apparent reason. This measure should therefore be complemented by other evaluation methods.

**The kernel density estimation :** this evaluation consists of fitting a distribution to the generated examples and finding the likelihood of the initial examples to belong to this distribution. We use the test set to generate behaviours from the audio features. These generations are then used to perform kernel-based density estimation. In this evaluation, each image is considered as a different sample. Finally, we compute the mean and standard deviation of the likelihood that the initial samples belong to this generated distribution. This measure gives a good indication of the reliability of our models, but cannot be used alone to evaluate them. One deviation may be greater than another and yet give more believable movements and expressions when they are visualised. We select models whose mean and standard deviation of likelihood is less than the average of all our evaluations.

**The visualisation from PCA reduction:** PCA reduces the visual dimensions of our samples, then we project onto a two-dimensional space several images from our initial test set as well as several images from our generated test set. A good distribution does not guarantee believable results, but a bad distribution generally reflects bad results. The 3 most frequent cases are : (1) distribution of generated data close to the distribution of real data; (2) distribution of generated data spatially shifted in comparison to the distribution of real data; (3) distribution of generated data centred on the distribution of real data, but reduced.

Based on the objective metrics described above, we select for each of our models a good architecture and combination of hyperparameters. Our first CGAN model is identified as "$m1$", our second AED model is identified as "$m2$" and finally this second model without its discriminator (corresponds to $w = 0$ in the loss function) as "$m2w/oD$". We first select models with the kernel density estimation whose mean and standard deviation of likelihood is less than the average of all our evaluations. Then, we keep those whose visualisation from PCA reduction showed the best distribution of generated data compared to the distribution of real data. For the best models, we obtain in terms of log-likelihood : mean -46.76, std 93.77 for $m1$; mean -51.36, std 101.692 for $m2$, and mean -51.35 std 97.24 for $m2w/oD$.

These three objective metrics, considered together, give an indication of the believability of the generated behaviours. However, they remain insufficient. For example, they do not measure the coherence of the behaviours with the speech or do not evaluate the adequacy of the behaviour speed. In general, objective measures are necessary, but not sufficient to determine which models give the best results [41]. Subjective evaluations are therefore crucial since the objective measures cannot assess all the complexity of the social communication. However, these studies are long and complex to implement, hence the use of objective metrics to pre-select models.

## Subjective Evaluation

The ultimate goal of behaviour generation is to generate behaviours that appear believable in comparison to human behaviours. Since human movements are highly variable, the generated movements may appear believable without matching the training data. Consequently, the best way to evaluate our models is to conduct user perceptive studies.

In order to select the appropriate evaluation criteria, we base our subjective evaluation study on previous research, such as Wolfert et al. [41], Wu et al. [42] and Habibie et al. [13]. We first selected two criteria : the naturalness and the temporal coordination with speech, to complement our objective measures. These two criteria are necessary to obtain a believable animation. We evaluate these criteria through direct questions:

- o naturalness: is the behaviour natural? Is the behaviour smooth?
- o temporal coordination: is the behaviour coherent with the speech? Is the speed of movements and facial expressions coherent with the speech?

In the conducted subjective evaluation, we randomly select seven videos among all videos of man from our test set and we choose from each of them a 15 second sequence. In order to simplify the animation process, the used virtual agent is always the same male character. We start by producing the animation videos of a virtual agent corresponding to the real videos. To do so, we use the visual features $\theta_{head}[0:T]$, $\theta_{gaze}[0:T]$ and $\theta_{AU}[0:T]$ extracted from each of the initial videos with OpenFace and animate the virtual agent on Greta with these features. The movements of the virtual agent are thus the movements performed by the speaker of the initial video. Note that, due to the limitation of Openface and of the Greta platform (limited number of AUs), the resulting video is not exactly a replication of the human's behaviour. We will therefore call these sequences the *simulated ground truth*.

Next, we associate the sound of the initial videos to our animated videos. To avoid the uncanny valley effect [27], and more particularly to avoid a gap between the realism of the voice and the realism of the virtual character, the pitch of the voice is modified to look like a synthesised voice[4]. We also choose to blur the mouth area. In our preliminary experiments, we noticed that the participants' evaluations were strongly linked to the accuracy of the lip movements. However, many tools exist to generate lip movements from speech, so we decide, in this first work, to direct the participants' attention to all other generated behaviours.

We repeat the animation process and replace the visual features of the *simulated ground truth* with those predicted by each of our models. In total, we animate 7 monologues from the test set, first with the real features extracted with OpenFace from the real videos[5], then with the features generated by our models $m1$[6], $m2$[7] and $m2w/oD$[8]. We obtain in total 28 videos of 15 seconds each.

Thirty-one persons of French nationality, recruited on social networks, participated to our study (16 males, 14 females and 1 not disclosed). The average age of the participants is 30.13 years with a standard deviation of 11.26. They viewed each of the videos, in a random order, and rated them on each of the criteria using a five-point Likert scale, ranging from strongly disagree (1) to strongly agree (5).

Table 2 presents the results of this objective evaluation for our three selected models and for the *simulated ground truth*. The values in the table are the means (mean) and standard deviations (std).

Table 2. Results of the perceptive study

|  | *Simulated ground truth* | | *m1* | | *m2* | | *m2w/oD* | |
|---|---|---|---|---|---|---|---|---|
|  | mean | std | mean | std | mean | std | mean | std |
| Coordination | 3,08 | 1,07 | 2,21 | 0,99 | 3,10 | 0,87 | 2,98 | 0,97 |
| Naturalness | 2,67 | 1,03 | 1.85 | 0.88 | 3,16 | 1,02 | 3,24 | 1,05 |

The average of the scores tends to show that the best model is the $m2$ model in terms of coordination, and the $m2w/oD$ model in terms of naturalness. We also note that for each model, the scores for coordination and naturalness are different, showing the importance to analyse these two criteria.

To further analyse the results, we perform a statistical analysis to assess the significant differences between the models. We conduct the Shapiro-Wilk test to assess the normality, which reveals that the data are not from a

---

[4]See the section *Add synthesised voice* on github.
[5]example: https://youtube.com/shorts/EKVDGSBY_wA?feature=share
[6]example: https://youtube.com/shorts/ytkPzso6l28?feature=share
[7]example: https://youtube.com/shorts/zJQrnR2mN4g?feature=share
[8]example: https://youtube.com/shorts/H9O9-k1pHx4?feature=share

normally distributed population. The evaluation is therefore performed using a Friedman test based on a repeated measures ANOVA, with the within-subjects factor being the considered model (ground-truth, $m1$, $m2$, $m2w/oD$).

Once again, the results show the superiority of the auto-encoding architecture ($m2$ and $m2w/oD$) which, by representing the acoustic speech features in a smaller representation and then decoding each of the visual features independently, allows to obtain results significantly superior to the $m1$ model in terms of naturalness, $p < 0.001$, and coordination, $p < 0.001$, but also to the *simulated ground truth* in terms of naturalness, $p < 0.001$. Secondly, none of the two criteria differ significantly between the $m2$ model and the $m2w/oD$ model. These results are different from what we expected. Indeed, we expect that the addition of the discriminator improve coordination, but the lack of significance in the difference between $m2$ and $m2w/Ow$ models does not allow us to draw such conclusions. These results may be explained by different reasons: (1) the discriminator avoids convergence towards the average and increases believable and expressive behaviours, as a result, they are perceived as less coordinated; (2) user tests performed on very short videos of 15 seconds, and a longer video could lead to different results revealing repetitive behaviours; (3) the number of participants may not be sufficient to reveal significant differences between these two models. Consequently, in the next evaluation, we aim at evaluating video with longer duration and on a larger set of participants.

## 7  CONCLUSIONS AND PERSPECTIVES

We present two models that jointly generate head movements, gaze orientation and facial expressions based on action units (AUs) automatically from speech, a Generative Adversarial Network and an Adversarial Encoder-Decoder. As far as we know, these models are the first attempt to generate jointly these non-verbal behaviours.

We implement a multi-step evaluation, first objectively with kernel density estimation and visualisation from PCA reduction, then subjectively through users subjective study, on two criteria: coordination with speech and naturalness. This evaluation shows the superiority of the model with an encoder-decoder. These results should, of course, be taken with caution, as a change in the length of the videos considered could for example change the participants' perception of certain criteria.

The proposed objective evaluation metrics allow us to differentiate models that generate completely unrealistic behaviours from those that generate more believable behaviours. To improve this evaluation phase, it is necessary to integrate new metrics able to compute the speed and coherence of behaviours with speech. Nevertheless, subjective evaluations are crucial in our field, mainly because social communication is much more complex than what objective measures are able to evaluate.

We implement during the subjective study a five-point Likert scale, which participants used to evaluate certain criteria such as the naturalness and temporal coordination of the virtual agent. In our next subjective studies, we could try to use pairwise comparisons. Participants will be asked to indicate which video between two proposed matches the evaluated criteria the most. We also would like to use longer generated videos and conduct the study with a larger number of participants. There are many other ways to improve this study, for example by using an eye tracker to assess participants' attention.

Despite the various experiments performed, there are still many possibilities to explore that would probably lead to a generation of more believable behaviours. We can add a regularisation term for the loss function or simply adapt the number of convolution layers of our models. The generated behaviours also strongly depend on the considered corpora. Our corpus lacks, among other things, of moments of "silence", so that our models cannot learn the behaviours to adopt when there is a pause in speech. In addition, the use of automatic extraction tools to extract behavioural characteristics add considerable noise to our dataset. Finally, the large number of

speakers in our dataset adds a difficulty to the learning process, we plan to replicate our experiments with a dataset containing a single speaker.

In our future work, several directions are possible. Firstly, we aim at generating behaviours conditioned by a social attitude. We are particularly interested in the persuasion. A possible metric in this context will be to use a generative model as a feature extractor, and then evaluate through a linear model its performance on the classification of the persuasive attitude [31]. Secondly, we aim at integrating the dyadic interaction environment, rather than just a monologue. An approach to simulate socio-emotional behaviour and to integrate the interaction would be to mix rule-based systems and data-driven approaches. This would allow us to take advantage of the benefits of both approaches simultaneously. Finally, instead of using human speech to generate the non-verbal behaviours, we could try to achieve this generation from a synthesised speech. The synthesised speech will probably have to present the same variations of these acoustic features as the human speech to obtain satisfactory results.

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
