# OpenReview forum: "Automatic facial expressions, gaze direction and head movements generation of a virtual agent"
_ACM.org/ICMI/2022/Workshop/GENEA — GENEA Challenge & Workshop 2022 Workshopproceeding_

### Official Review · Reviewer_bC8y · 2022-08-15
**The paper presents two architectures to generate head movement and facial expressions from speech.**

**Rating:** 4
**Confidence:** 4

**Review:**


The authors use OpenFace to extract Action Units. The extracted data can be quite noisy. The authors do not provide information on the data processing they applied. The authors also do not say which AUs they consider. In the videos, the mouth is blurred. In the 'smoothing of the data' section, why the authors only smooth the eyebrow AUs and not the others?
How gaze is defined? Is it defined as eye direction? How is it detected? OpenFace does not distinguish between head and eye movement.
In the objective study, the authors compare their results with those of Sadoughi and Busso. But Sadoughi and Busso do not model facial expression. So how meaningful is such a comparison?

The subjective evaluation is weak. Only 7 stimuli from each condition were used. It is a too small number that does not allow having reliable results.
How long were the stimuli? Where there any attention checks?
How the participants were recruited?
The stimuli are in English but the participants are French. Could it create bias  in the perception of the animations?

The animations of the agent are not very good: the head is tossing quite a lot, making sequences of rather similar movements. The eyebrows are flickering. There is not an eyebrow action with a higher amplitude that would mark a pitch accent.

There are no comparisons with the previous models. So it is not possible to judge if the presented model is better than the state of the art.
The paper has several details missing which will make replication of work very hard.

---

### Official Review · Reviewer_QTWx · 2022-08-18
**Great contribution**

**Rating:** 9
**Confidence:** 4

**Review:**

In this paper, the authors build and compare two different ML models for jointly generating head, gaze, and facial movements from an input speech signal (acoustic), including a GAN and an adversarial encoder-decoder. Data from the CMU Multimodal Opinion level Sentiment Intensity corpus is used for training and testing the models. Face/head features are extracted with OpenFace, and speech features are extracted with OpenSmile. The models are evaluated both objectively (examining the loss functions, kernel destiny estimation, and PCA reduction) and subjectively (in a user study with 31 people). The encoder-decoder architecture was judged to perform best in terms of perceived naturalness.

This paper represents a very impressive piece of work, and it will be a great fit for the workshop. The methods are described very clearly, and the results are convincing. I particularly appreciated the structure employed when presenting the relevant related work and state of the art, which was very informative.

A few questions and comments:
I did not fully understand the need for the "smoothing of data" presented at the end of Section 5. The authors state that "The speed of the generated behaviours is higher than real behaviours." But why was that the case? What is the intuition behind this? Why would the generated speeds be any different from the statistical distribution of the dataset used to train the models? The authors should also provide more justification for why the eyebrow AUs need to be smoothed, but not the other ones.

I would caution against the characterization that some of the generated behaviors were perceived as "more natural than the ground-truth," which is a quite suspicious statement to make in the first place. The authors do acknowledge some important caveats about the ground truth, which is not completely faithful in reproducing human behavior due to limitations in the platform and tools used. Combined with the limitations in evaluation (e.g., the length of the video clips that were evaluated without any other context), I would suggest entirely avoiding any claims about the model performing "better" than ground truth.

One suggestion I have when deciding which directions to pursue in future work, is to be guided by the potential applications and scenarios for these sorts of models. Interactive autonomous virtual agents immediately come to mind. But such agents will undoubtedly require speech synthesis in addition to generated nonverbal behaviors. How well would these models work when driven by *synthesized* speech, rather than recordings of natural speech?

**Nominate For A Reproducibility Award:**

This work is potentially deserving of the reproducibility award. An openly available dataset was used, and all tools used were open source. They have released all code on GitHub with fully detailed procedures.

---

### Official Review · Reviewer_tdHb · 2022-08-18
**AI-driven animation with dubious validity**

**Rating:** 5
**Confidence:** 5

**Review:**

This paper describes experiments with generation of the movements of an avatar using two different ML-models - one probabilistic (GAN) and one deterministic. The models are trained using data from different people, recorded under varying conditions, that have been tracked using OpenFace and then evaluated according to some objective metrics and through a user study.
The paper is clear and easy to follow, and the machine learning models are sensible for the intended purpose. Source code is provided which is a big plus.
However, the validity of the experiments are dubious, for a number of reasons. First of all, the training data comes from 89 different speakers recorded under different conditions, with an average of under 3 mins per speaker, which are then used to train one model. Most likely this is too little and diverse data for any model to learn consistent behaviours? Also it is generally a bad practice to train a behaviour generation model on multiple people because at best the models will learn to produce an oversmoothed average behaviour which doesn't represent the behaviour of a real person. It is also not specified in the paper if the test/train division is done on a per-speaker basis so that no same person appears in both sets.

Looking at the provided video clip of ground truth makes me question the quality of the training data. It appears choppy and noisy, which most likely is a consequence of home-webcam quality and the openface tracking used. In combination with the multi-speaker nature of the data this must make a very hard task for the networks to learn. With that in mind it is unsurprising that heavy post-smoothing is required to produce the final videos.

In the subjective evaluation, subjects were asked "is the behaviour natural? is the behaviour smooth?". First, these are two rather different things, Natural behavior is not always smooth, but secondly, since the ML-generated movements were explicitly smoothed whereas the GT was not (?), there is no surprise there either that the M2 is prefered over GT.

I also severely question the decision to smooth the eye movements with a savitzky-golay filter - saccades un-smooth by nature so smoothing gaze will effectively make it very un-natural. I believe this is evident also from the provided examples - in the ground truth, the agent is looking at the camera for the most part, whereas in the models, gaze just follows wherever the head points, which is again not what a real person would do (in fact it is very difficult to move your head without fixating your gaze on anything).

In summary, the pros of the paper is that it is well written and compares both probabilistic and deterministic methods, and includes a subjective evaluation and source code. The downside is that it is impossible to draw any conclusions from these experiments since my feeling is that the model outputs are largely random, which ultimately seems to go back to the training data.

---

### Decision · Program_Chairs · 2022-08-20

**Decision:**

Accept (Workshop proceeding)

**Comment:**

This paper got three extensive reviews. One of the reviewers places questions at how the dataset was used, and the quality of the dataset. Another reviewer asked how certain terms are defined, such as gaze, as this is not defined in the paper itself. There is also no comparison with previous models, but only with the ground truth. The resulting animations are considered not very good by two of the three reviewers. One reviewer is really fond of this work and places additional questions which should be addressed by the authors in their work.

We suggest that the authors carefully consider the feedback received from the reviewers and use it to improve their manuscript for the camera-ready submission.